# Early Maladaptive Schemas and Schema Modes among People with Histories of Suicidality and the Possibility of a Universal Pattern: A Systematic Review

**DOI:** 10.3390/brainsci13081216

**Published:** 2023-08-17

**Authors:** Anna Grażka, Dominik Strzelecki

**Affiliations:** Department of Affective and Psychotic Disorders, Medical University of Lodz, 92-216 Lodz, Poland; annagrazka.psycholog@gmail.com

**Keywords:** early maladaptive schema (EMS), schema mode, suicidal ideation, suicidal attempts, systematic review

## Abstract

Background: The identification of variables affecting suicidality and the search for interventions to reduce suicide risk are priorities among mental health researchers. A promising direction for such research is schema therapy and its two main constructs, i.e., early maladaptive schemas (EMSs) and schema modes. Methods: This systematic review was designed in accordance with the PRISMA guidelines. It summarizes the studies conducted to date that describe the relationship between EMSs and schema modes and measures of suicidality in individuals over the age of 16. Results: The review confirmed that there are many significant associations between EMSs (especially from the Disconnection/Rejection domain) and suicide risk. Although only one study was found that explores the association between schema modes and suicidality, the correlations it identified are also confirmed here. Discussion: The results show the unquestionable importance of EMSs and schema modes in assessing suicide risk. The co-occurrence of these variables represents the first step in further assessing causality and introducing schema therapy techniques into work with patients who are at risk of suicide. This issue requires more extensive experimental research.

## 1. Introduction

From the beginning of our lives, in our interactions with both significant others and the world, we undergo experiences that condition our development. Young et al. [1] distinguished five categories of universal basic early childhood needs, the satisfaction of which allows an individual to develop optimally. These needs include (I) secure attachment, (II) autonomy, a sense of competence, and identity, (III) the freedom to express needs and emotions, (VI) spontaneity and play, and (V) the need for realistic boundaries and self-control. When these needs are not met, the so-called early maladaptive schemas (EMSs) develop. An EMS—one of the main concepts in schema therapy—is defined by Young as “a broad, pervasive theme or pattern, comprised of memories, emotions, cognitions, and bodily sensations, regarding oneself and one’s relationships with others, developed during childhood or adolescence, elaborated throughout one’s lifetime and dysfunctional to a significant degree” [1]. Young identified 18 ESMs and assigned them to five domains, as briefly described in Table 1.

In contrast to EMSs, viewed as traits, schema modes have also been distinguished. “Modes refer to the predominant emotional state, schemas, and coping reactions that are active for an individual at a particular time” [2]. This construct describes a state consisting of activated EMSs and associated coping reactions in response to a stimulus recalling the emergence of a schema. Schema modes can take the form of adaptive and maladaptive responses to a situation (involving emotional, cognitive and behavioral aspects) in which an individual’s needs are not fulfilled. Table 2 describes the 14 schema modes identified by Lobbesteal et al. [3].

Numerous studies prove that EMSs and dysfunctional schema modes are more common in patients diagnosed with various psychiatric disorders. A higher prevalence of EMSs has been confirmed in patients with depression [4,5], anxiety disorders [6,7], obsessive–compulsive disorder (OCD) [6], bipolar disorder [8,9,10], schizophrenia [11], eating disorders [12], substance abuse [13], and personality disorders [14,15]. The presence of EMSs has also been linked to impaired functioning of individuals, a more severe course, and worse treatment outcomes in patients diagnosed with affective disorders [5,9,10]. Increasing number of studies also show the impact of EMSs on suicide risk. Most often, the studies involve a diagnosis-selected group of patients. It can result in overlapping schemas specific to a particular disorder and to suicide risk. For this reason, some studies have begun to control for symptom severity [16,17,18,19].

From the perspective of mental health professionals, the health care system as a whole, and also of families or friends, and, above all, of those at risk of suicide themselves, it is crucial to determine the factors that influence suicidality. EMSs and schema modes seem to be a good direction of exploration, clearly connecting with existing theories of suicide.

According to the Interpersonal-Psychological Theory of Suicidal Behavior (IPT) by Joiner et al. [20], individuals at suicide risk are characterized by high levels of perceived burdensomeness (the feeling of being a burden to family, friends, or society) and failed belongingness (the experience of being alienated from others). Failure to meet early childhood needs with high confidence can result in similar feelings. In another model, the Three-Step Theory (3ST) by Klonsky and May [21] posits that suicide (I) is the result of a combination of pain and hopelessness, (II) is contrary to one’s sense of connectedness to others, and (III) requires a progression from ideation to attempts through acquired contributors to the capacity to attempt suicide such as fearlessness about death and increased physical pain tolerance. Furthermore, the third Integrated Motivational and Volitional Model of Suicidal Behavior (IMV) by O’Connor et al. [22] predicts that suicide occurs in a three-phase process. The pre-motivation phase (I) includes background factors (i.e., an individual’s biological, genetic, and cognitive vulnerability with social and environmental context) and triggering events. The motivational phase (II) begins with feelings of failure and humiliation, leading to a sense of entrapment and ultimately to the emergence of suicidal thoughts. And the volitional phase (III) involves the transition from suicidal ideation to enaction with the help of components of the acquired capability, but also variables of an environmental, social, psychological, or physiological nature.

All of the above theories are categorized as second-generation models because they treat suicide as a process, with the ideation-to-action framework [23], in contrast to older models that did not distinguish between thoughts, tendencies, and readiness to act. They indicate that a presence of a mental disorder diagnosis may be one of many triggers, but central to this is the configuration of numerous motivational and volitional variables that vary at different stages of the process leading to a suicidal act. Looking at schema theory, its constructs like EMSs, and perhaps even more schema modes (involving a behavioral factor) seem to be an accurate predictor that includes both the trigger stimulus, the cognitive part, and the behavior that can lead to suicide.

### The Current Review (Objectives)

The aim of the current systematic review was to clarify what EMSs and schema modes are present among late adolescents and adults with a history of suicide. We wanted to analyze whether it is possible to establish a universal pattern of EMSs or schema modes among people with a history of suicidal thoughts, tendencies, and attempts, or whether this is an issue specific to the disorders they suffer from (dependent on diagnosis). We also intended to summarize current reports in the context of knowledge about EMSs and schema modes during the period of psychological stability compared to suicidal crisis.

In designing this review, we go beyond the implications made in the article by Pilkington et al. [24], who examined the correlation of EMSs’ role with suicidal ideation and self-harm behavior. They pointed to the likelihood that schema modes are the more relevant variable in predicting suicidality. Furthermore, in our review, we want to expand the analysis of the association of EMSs with current suicidal ideation by including EMSs specific to suicidal tendencies and behavior. This is in line with the postulates of the second-generation models of suicide presented above, treating it as a process. As such, we assume that different constellations of EMSs may be at work at various stages of this process. Moreover, our review focuses exclusively on suicidal thoughts, tendencies, and behaviors, leaving out the topic of self-harm, given its different purpose: suicidal or non-suicidal [25].

Conclusions drawn from the abovementioned goals may have broad practical applications in the context of suicide risk assessment in both outpatients and inpatients. Determining the EMSs and schema modes most closely associated with suicide risk would also allow increased attention to them in psychotherapeutic practices as factors sustaining a suicide crisis. The identified EMSs or schema modes could then be a part of the primary focus within ongoing therapeutic interventions. Finally, the results obtained, can serve as a direction for further research by pointing out missing data or weaknesses in the studies analyzed.

## 2. Methods and Design

The presented systematic review on EMSs and schema modes in suicidal individuals was prepared in accordance with the Preferred Reporting Systematic Reviews and Meta-Analyzes (PRISMA) guidelines (see: Appendix A). The protocol was registered in the PROSPERO database of systematic reviews (https://www.crd.york.ac.uk/prospero (accessed on 20 January 2023); registration number: CRD42023383048). 

### 2.1. Eligibility Criteria

The following are the inclusion criteria for studies in the review: (I) a sample consisting of late adolescents and adults (16 to 70 years old); (II) assessment of at least one of the following variables: suicidal thoughts, suicidal tendencies/plans, suicide attempts, and suicide risk; (III) assessment of EMSs or schema modes; (IV) observational studies (in particular: cross-sectional studies, case-control studies, cohort studies, longitudinal studies); and (V) published peer-reviewed English- and German-language articles.

We excluded the following studies: (I) children and early adolescents (under 16 years old) and older population (over 70); (II) individuals diagnosed with intellectual disability and dementia syndromes; and (III) review papers, meta-analyses, case report studies, book chapters, discussion papers, and gray literature.

The exclusion of children and early adolescents from the study was related to the fact of the different specificity of this group compared to the study group (e.g., what in the case of a child or early adolescent would be a normal reaction, in the case of an adult would be an expression of the Impulsive Child mode). The upper age limit, on the other hand, indicates that studies involving people over 70 are lacking.

Figure 1 illustrates a flowchart of selecting studies for this systematic review. A detailed description of the research selection process is included in Appendix A. 

Studies were grouped for analysis considering the patients’ diagnosis, the presence of an assessment of EMSs or schema modes, and the type of suicide assessment (thoughts, tendencies, attempts, or risk).

### 2.2. Literature Search Strategy

First, we established a search strategy, identifying keywords and their combinations. The result was the following equations: (“early maladaptive schema” OR “maladaptive schema” OR “schema mode”) AND (“suicide” OR “suicidal risk” OR “suicidal ideation” OR “suicidal attempt” OR “suicidality”),(“Frühe Maladaptive Schemata” OR “Maladaptive Schemata” OR “Schemamodus”) AND (“Selbstmord” OR “Suicidrisiko” OR “Selbstmordgedanken” OR “Selbstmordversuch”)

Which we adapted to the requirements of each base. We limited the search to peer-reviewed articles on people aged 16 and older.

The source search was implemented until the end of March 2023. We used five electronic databases in the search: PubMed, EBSCO, ScienceDirect, Scopus, and The Cochrane Library. The literature of studies meeting the inclusion criteria was manually searched to identify additional eligible studies.

The search was repeated before the final analysis. The authors checked all databases and collected data separately. Discussion and agreement by both investigators preceded the inclusion of a study in the list.

### 2.3. Methodological Quality Assessment

Only peer-reviewed studies were selected for analysis. In addition, to limit the risk of bias, both researchers independently assessed the selected studies using the Newcastle–Ottawa Quality Assessment Scale adapted for cross-sectional studies by Modesti et al. [26]. The scale assessed three areas: (I) selection (representativeness of the sample, sample size, response rate, and ascertainment of exposure), (II) comparability of subjects in groups and control of confounding factors, and (III) outcome (assessment measures and correct application of the statistical tests). Differences in scores resolved via consensus and the final selected studies received at least seven points, entitling them to be classified as good or very good cross-sectional studies.

### 2.4. Main Outcomes

The review aimed to determine the association between specific ESMs and specific schema modes in patients with a suicidal ideation, tendencies, and history of suicide attempts in late adolescence and adulthood.

The outcomes assessed had to be measured using standardized and validated measures of EMS and schema modes (e.g., Young Schema Questionnaire and Schema Mode Inventory). The effect measure was assessed mainly by determining the mean difference between subgroups or occurrence of correlations between variables.

### 2.5. Data Extractions

We obtained data from studies that met the inclusion criteria with the help of extraction sheets and a codebook. Both authors independently searched each research to minimize the risk of erroneous or inaccurate data extraction.

From studies meeting the full inclusion criteria, we extracted the following data: (I) reference information about the study (author, year, location); (II) the objectives of the study; (III) basic sample data (sample size and characteristics of participants); (IV) assessment measures; and (V) main results and conclusions. We found the listed information in each selected study.

### 2.6. Data Synthesis

Given the specifics of the research described, we used a narrative synthesis. We conducted a preliminary synthesis to organize the data. The data were then extracted and presented in tabular form to facilitate visualization, observation of relationships within and between studies, and comparisons. 

In addition, we created a textual description of each included study along with a critical evaluation of the study. This was followed by an analysis of the similarities and differences between each study using a clustering technique.

Finally, we critically reviewed the results, considering the strengths and limitations of each study. We clarified any doubts and discrepancies through discussion.

## 3. Results

### 3.1. Included Studies

We searched five electronic databases, resulting in 79 articles (19 from PubMed, 26 from EBSCO, 12 from ScienceDirect, 19 from SCOPUS, and three from The Cochrane Library). After removing duplicates, 42 articles remained whose titles were analyzed. We eliminated 27 because they did not meet the inclusion criteria (most were unrelated to the topic, dealing with younger adolescents or non-suicidal self-injuries). We analyzed the remaining 15 articles by reviewing the abstracts. At this stage, we excluded three manuscripts (one without EMSs or schema modes measures and two concerning younger adolescents). After manually searching references, we found that none met the inclusion criteria. Finally, we included 12 studies in the review. We showed the general characteristics of the selected studies in Table 3.

### 3.2. Study’s Characteristics

The articles included in the review were published between 2008 [27] and 2022 [34]. Their highest number is from 2017 (n = 4). The study was conducted with subjects from three continents: five from Asia [16,17,18,31,34], five from Europe [19,28,29,30,33], and two from North America [27,32]. All the manuscripts were from peer-reviewed scientific journals.

### 3.3. Sample’s Characteristics

The sample size of the selected studies ranged from 49 [30] to 766 [32] participants. The mean age of the subjects ranged from 19.9 [32] to 45.4 [31]. The study group consisted of both psychiatric treated and non-treated individuals. In some of the analyzed groups, the participants represented a specific diagnosis (e.g., depressive disorder [19,31], borderline personality disorder [29,33], bipolar disorder [18], obsessive–compulsive disorder [17], schizophrenia [16]), and other studies involved individuals seeking help for experienced symptoms [32,34] or cross-sectionally studied students [32,34]. 

Most of the studies compared a group at high suicide risk with a group with the same diagnosis at lower risk. Two studies also included a non-clinical control group [28,31]. Three studies did not select a control group [27,32,33].

### 3.4. Measurement Characteristics

Standardized and validated self-report questionnaires were used in the studies analyzed. EMSs were mostly tested using a shortened version of Young’s Schema Questionnaire, composed of 15 subscales. One study used a modified version of the shortened scale [27], which measured 12 subscales, while four used scales that measured all 18 EMSs [19,29,30,31]. Schema modes were measured in only one study [29].

There were significant discrepancies between the instruments used to measure suicide risk variables. The instruments examined the following: the presence of suicidal ideation, suicidal plans, suicide risk, and the history of past suicidal attempts. Some studies used different self-report questionnaires for this purpose e.g., [33,34], while others relied on data from medical records and interviews with subjects e.g., [30,31].

Other measures assessed the severity of symptoms of comorbid disorders associated with suicidal crisis and acted as control variables in some studies [16,17,18]. 

All studies used a cross-sectional design.

### 3.5. Main Findings

Table 4 summarizes the relationships between suicide risk as expressed by various measures (including control variables) and EMSs.

#### 3.5.1. EMSs in People at Suicide Risk

Analyzing the results presented, it is explicit that the domain most closely associated with suicide risk is Disconnection/Rejection. In this area, the individual anticipates that he cannot count on the gratification of basic emotional needs such as security, stability, care, empathy, unconditional acceptance, or respect [1]. Five studies [16,18,31,33,34] proved that all the EMSs identified in this domain in patients with high suicide risk consider high severity scores. Several studies have demonstrated significant differences between the mean severity scores of EMSs from Disconnection/Rejection area in patients at suicide risk compared to controls [19,29,30,31]. Others, however, examined the measurement of correlations between suicide risk scores and EMSs severity from this domain [16,17,27,28,32,34]. One study obtained data of both types [18], and one used regression analysis [33]

It is noteworthy that schemas from domain Disconnection/Rejection are associated with suicide risk and assessed either by suicide ideations, plans, risk, or history of attempts. These relationships also hold when controlling for variables such as the level of severity of depressive [16,17,18], anxiety [17], or manic symptoms [16]. 

The schemas most strongly associated with suicidality confirmed in almost all included studies were the Social isolation/Alienation and Defectiveness/Shame schema. Several studies [17,18,28] also highlighted the effect size (medium to large) for the two schemas indicated. The common factor in these two schemas is a feeling of being unworthy of inclusion in society by one’s dissimilarity or defectiveness. 

The data obtained are consistent with the Interpersonal-Psychological Theory of Suicidal Behavior (IPT) by Joiner et al. [20], according to which high levels of perceived burdensomeness and failed belongingness predispose individuals to commit suicide. Similarly, the Three-Step Theory (3ST) by Klonsky and May [21] recognizes that suicide conflicts with a sense of connectedness with others. The findings are also consistent with the outcomes of the meta-analysis on risk factors for suicide in the general population by Favril et al. [35]. The results we considered were obtained using the psychological autopsy method, which involved collecting information from proxy informants and (when available) reviewing medical and coroner’s records. Social isolation appeared to be the most significant risk factor in the area containing sociodemographic variables. Similar conclusions reached Saeed et al. [36], who compiled a meta-analysis of studies on suicide during the COVID-19 pandemic. Loneliness was cited as one of the strongest predictors of suicidal behavior, and social connection was found to be a crucial preventive variable [37].

Another associated schema for suicidal behavior revealed the Emotional deprivation schema. Individuals representing this schema show anticipation that their desire for emotional support will never be satisfied by others. This schema may be expressed in the fear of lack of care (i.e., interest, affection, companionship), empathy (i.e., understanding, being listened to), or protection (i.e., guidance or direction from others) [1]. A relationship with suicidality at a similar level was observed between the Abandonment/Instability and Mistrust/Abuse schema. The first schema is related to the perception of significant people as unstable and unpredictable, in relation to whom there is doubt that they can be relied upon. Characteristic is the fear of losing them (through illness, death, or abandonment for someone else). The second schema is related to the belief that others are a threat. They intentionally hurt, humiliate, cheat, or take advantage [1]. The three schemas discussed above often develop as a result of experiencing physical violence, emotional abuse, or neglect that occurs during childhood. The link between such experiences and suicide risk has been confirmed in a number of studies, clustered in a meta-analysis by Norman et al. [38]. 

The second most active area of EMSs in individuals at high suicide risk is Impaired autonomy/performance. This area includes schemas of belittling one’s ability to survive, function independently, and operate efficiently. Among the other areas most strongly associated with suicide risk was the Dependence/Incompetence schema. People with an increased presence of this schema have the conviction that they lack the ability to handle daily responsibilities without the help of others. As a result, they often feel helpless [14]. The review also proved links between suicide risk and the Vulnerability to harm and illness and Failure schema. The first involves experiencing an exaggerated fear of imminent disaster in the sphere of physical health, mental health, or the presence of an external event (e.g., earthquake or plane crash). The second consists of an individual’s belief that they have failed and will undoubtedly fail in various areas of achievement [1].

Similar conclusions were reached in a systematic review by Costanza et al. [39] on the demoralization phenomenon, understood as a distress syndrome resulting from a sense of subjective incompetence [40]. The authors provide data showing that demoralization is a significant factor that increases the risk of suicidal ideations and behavior.

We noted significantly fewer correlations and smaller effect sizes between measures of suicide risk and EMSs from another domain, i.e., Impaired limits. This domain contains EMSs that cause difficulties in respecting the rights of others, cooperating with others, and establishing and pursuing realistic goals [1]. Significant correlations were reported in this area without controlling for other variables that may influence suicide risk. When researchers controlled for depression levels, for example, they did not obtain significant results [16]. It is noteworthy that an association between suicidal measures and the Entitlement/Grandiosity schema (related with the belief that one is superior to others and entitled to special rights and privileges) emerged in a group of patients with a diagnosis of bipolar disorders (BP) [18,30]. Admittedly, the patients were tested in remission, but according to the systematic review by Bär et al. [41], individuals with BP present stably higher levels of EMSs activation (associated with depressive symptomatology as well as with manic symptomatology) relative to both clinical and non-clinical controls.

The Other directedness domain (related to attaching excessive focus to the needs and feelings of others) showed few associations with suicide risk. The researchers found similar results regarding the Hyper vigilance/Inhibition domain (schemas involving excessive emphasis on repressing spontaneous impulses and feelings). On the one hand, the small number of associations may be due to data deficiencies, as only four studies [19,29,30,31] used the full version of the Young Schema Questionnaire, which contains 18 EMSs. On the other hand, it is worth looking at the characteristics of the schemas included in these domains. Note that five of them (i.e., Subjugation, Self-sacrifice, Approval seeking/Recognition seeking, Emotional inhibition, and Unrelenting standards/hypercriticalness) were assigned to the group of so-called conditional schemas by Jeffrey Young. 

According to Young et al. [1], conditional schemas are often the individual’s way of coping with more primary, unconditional schemas. They enable the individual to take control (e.g., make sacrifices, suppress emotions, strive to meet high standards) to reverse a bad scenario. This is not a long-term solution, but it at least temporarily protects against experiencing the discomfort of activating unconditional schemas. In turn, unconditional schemas are the most at the core, formed earliest, expressed in the individual’s beliefs about himself and others. They leave the individual with no hope, regardless of his actions. 

Given the above, conditional schemas provide hope for dealing with unmet needs, in contrast to the feeling of hopelessness, considered one of the most important predictors of suicidal thoughts, suicide attempts, and death i.e., [21,42,43,44]. It has been proven in numerous studies that reasons for living and hope provide a significant protective effect [45].

#### 3.5.2. Schema Modes in Patients at Suicide Risk

Only Leppänen et al. [29] compared schema modes in patients with borderline personality disorders presenting parasuicidal behavior and in patients without such behaviors. The findings indicate that there are differences in terms of the presence of both dysfunctional modes and healthy modes. The group with parasuicidal behavior had significantly higher scores for four modes, including those of two from the inner child category: the Vulnerable Child and the Angry Child, and two from the coping mode category: the Detached Protector and the Compliant Surrender. At the same time, this group was characterized by a lower intensity of functional modes, i.e., the Healthy Adult and the Happy Child modes. 

The results obtained are consistent with the assumptions of schema mode theory. Dysfunctional child modes are an individual’s response to unmet basic emotional needs, leading, in the case of the Vulnerable Child mode, to feelings of sadness, hopelessness, fearfulness, overwhelm, or helplessness, and, in the case of the Angry Child mode, to feelings of anger or rage [46]. Rafaeli et al. [2] consider child modes as the most explicit and unambiguous manifestation of the activation of EMSs. Experiencing vulnerability and distress causes patients high levels of emotional pain, so they try to avoid being in dysfunctional child modes by using coping modes. This is a mechanism confirmed in the cited study. It proved that patients with BPD and suicidal behavior are more likely to use Detached Protector and Complied Surrender modes. The first involves emotional avoidance, resulting in the individual remaining detached, numb, and overly rational. The second expresses in excessive subordination, whereby the individual attempts to fit in with the expectations or demands of others at the expense of self. Both modes only deepen the discomfort as they still keep the individual unable to meet their needs [1].

In contrast, a strong Healthy Adult mode includes efficient thought processes and behaviors necessary for an adult to correctly perform tasks such as working, parenting, accepting responsibility, engaging in interpersonal relationships, and other activities. The mode also creates space for pleasurable activities such as intellectual, cultural, aesthetic interests, sex, health care, or sports. If this mode works well, it establishes the area of freedom necessary to develop the Happy Child mode. In this mode, the individual is calm, safe, cared for, and feels a connection with other people. Because he has a sense of fulfillment and worth, he can indulge in optimism, spontaneity, and feelings of contentment [2].

#### 3.5.3. EMSs in Suicidal Patients Depending on the Presence of a Psychiatric Diagnosis

By analyzing the studies included in the review, it was evident that those with a psychiatric diagnosis compared to those without, scored significantly higher on the assessed EMSs [28,31,32]. Moreover, those with higher suicide risk had the highest EMSs severity, as confirmed in several studies [16,17,18,27,28,31]. That is consistent with a number of sources i.e., [4,5,6,7,8,9,10,11,12,13,14,15,47,48,49], according to which EMSs are recognized as a significant correlate underlying many mental health problems. The present results further show that suicidal crisis is a highly severe symptom that exacerbates the presence of already active EMSs in patients with psychiatric diagnoses. This phenomenon is evident when controlling for variables such as severity of depression, anxiety, or OCD [16,17,18].

## 4. Strengths and Limitations

This systematic review provides valuable information on EMSs and schema modes found in patients at suicide risk. The results received can provide a framework for designating the most vulnerable to suicide risk EMSs. The review has been prepared following the Preferred Reporting Items for Systematic Reviews and Meta-Analyses (PRISMA). It is based on 12 studies conducted on different populations living on different continents. The analysis undertaken includes both untreated and treated psychiatric patients with various diagnoses.

The obtained outcomes, however, should be treated with some caution. The first limitation is because of the relatively small number of studies included in the review, suggesting the need for further exploration of the issue described. Secondly, the studies were conducted in different cultural contexts, which may have made it difficult to generalize the results due to the cultural determinants of suicide, as reported by Khosravani et al. [17]. Third, the extracted studies relied mainly on self-report measures. That may have affected the lower reliability of the data obtained. Another limitation was undoubtedly the use of a cross-sectional research design. It makes it difficult to assess the phenomenon over time and establish causality, so there is a need for longitudinal studies.

Also, complicating the analysis was the authors’ use of different measures of suicidality. We know from suicide theory [21] that suicidal ideations, tendencies, and behaviors are treated as part of a process, and each has its own set of explanations and risk factors. Available data are insufficient to establish any pattern within these categories. In future studies, it would be valuable to establish a homogeneous measurement to obtain the most accurate comparisons. A similar issue applies to the EMSs measurement. Due to the application of different versions of the Young Schema Questionnaire (shortened and comprehensive), most studies received results on a limited number of schemas. Moreover, EMSs have been assessed at different intervals since the suicide crisis. Given that they are treated as traits that may or may not manifest themselves depending on the individual’s emotional state, the reliability of measuring them at any time is questionable. Perhaps schema modes, reflecting an individual’s current response to circumstances, are the more appropriate construct, but we have too little data to confirm it. Finally, only a few studies used comparisons controlling for co-occurring variables. There is a need to expand such studies, including controlling for co-occurring variables, especially personality disorders, which were not presented in any of the papers.

## 5. Conclusions: Future Research

The main objective of this systematic review was to summarize the existing knowledge on the association between EMSs or schema modes and suicide risk in late adolescents and adults. The majority of the studies analyzed confirmed this association. Patients with higher suicide risk presented higher severity of most EMSs compared to patients without psychiatric diagnosis and those suffering from various mental disorders who did not experience a suicide crisis. The most significant differences in the severity of EMSs were noted in the Disconnection/Rejection domain, especially the Social isolation/Alienation and Defectiveness/Shame schema. Considering the schema modes, patients with suicide risk showed the highest intensity in two child modes, i.e., Vulnerable Child and Angry Child, and in two modes from the category of coping strategies, i.e., Detached Protector and Compliant Surrender. The above findings were also confirmed when controlling for the severity of co-occurring symptoms such as depression, anxiety, OCD symptom severity, and mania symptoms. This provides preliminary support for the hypothesis that a universal, diagnosis-independent pattern of EMSs exist in patients at suicide risk.

The analysis also found EMSs marginally associated with suicide risk. These mostly turned out to be the conditional schemas identified by Young et al. [1]. Due to the use of heterogeneous measures of EMSs, this issue requires further exploration. Considering the schema modes, the protective effects of the Healthy Adult and Happy Child modes have been proven.

The data obtained (especially on the relationship with schema modes) based on a limited number of studies restricted to specific patient groups, so broader studies with a more representative sample are needed. It would also be recommended to conduct a longitudinal study to check the stability of the results obtained, comparing the profile of EMSs during a period of relative stabilization of the mental state and in a suicidal crisis. Admittedly, EMSs are considered traits [2], which would suggest their permanence over time. However, it is known that a stimulus recalling the event that led to its formation is needed to activate EMSs. According to the authors, a better measure seems to be Schema Modes, as constructs reflecting the self-concept manifested at a particular moment. This issue requires further research, as no answer has been found.

In conclusion, the review provides information on the most common EMSs and schema modes in suicidal patients. This knowledge could be applied already at the diagnostic stage as part of the suicide risk assessment. It would be a method that does not explicitly test for suicide risk (unlike most techniques), which the person being assessed might feel more comfortable with. After all, sometimes it is difficult for the respondent to admit directly to suicidal thoughts or tendencies due to fear of being evaluated or of other consequences that might follow (such as involuntary hospitalization).

Awareness of EMSs and schema modes associated with suicide risk can also be helpful at the treatment stage. Admittedly, this review focuses on the co-occurrence of the schema therapy constructs and suicidality, but nevertheless, it could be a prelude to an expanded examination from the causality side. Confirmation of the contribution of ESM and schema modes to suicide risk would provide immediate tools for helping patients, resulting in more effective treatment and its reduced length. This would be beneficial both from the perspective of the patient and the health care system, reducing the costs of the interventions carried out.

## Figures and Tables

**Figure 1 brainsci-13-01216-f001:**
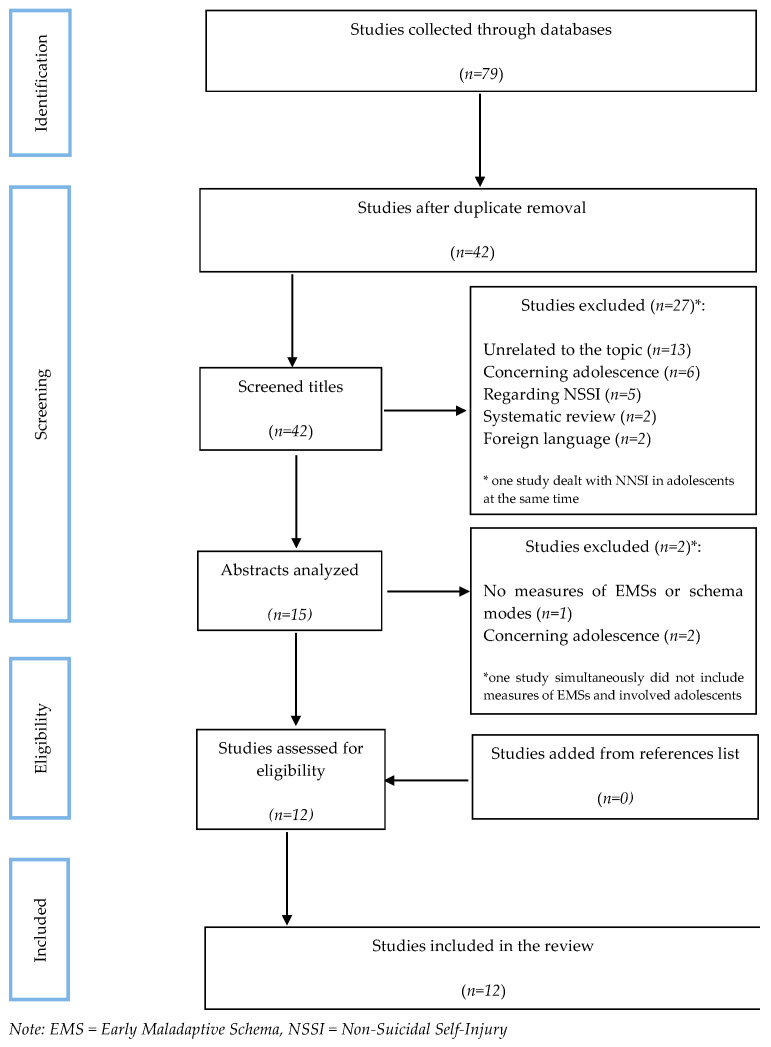
PRISMA (Preferred Reporting Items for Systematic Reviews and Meta-Analyses) flow chart of selection of studies.

**Table 1 brainsci-13-01216-t001:** Description of the 18 EMSs grouped into domains.

Domain	Early Maladaptive Schema	Description
Disconnection/Rejection	Abandonment/instability	The perception that close relationships are unstable and can be lost at any time.
	Mistrust/abuse	The conviction that others will intentionally hurt, humiliate, cheat, or take advantage.
	Emotional deprivation	The conviction that the desire to receive a normal level of emotional support from others will not be sufficiently satisfied.
	Defectiveness/shame	A sense of incompleteness, of being bad, unwanted, or inferior, which makes it impossible to be loved and accepted.
	Social isolation/alienation	The belief that one is isolated from the world, different from other people, and does not belong in society.
Impaired autonomy/performance	Dependence/incompetence	A sense of one’s inability to cope with daily responsibilities without significant help from others.
	Vulnerability to harm and illness	Feeling an exaggerated fear of the occurrence of a disaster (health, emotional, external) that cannot be prevented.
	Enmeshment/undeveloped self	Excessive emotional involvement and closeness with at least one person, leading to the impairment of full individuation or normal social development.
	Failure	An individual’s belief that he or she has failed and is incompetent compared to others in areas of achievement.
Impaired limits	Entitlement/grandiosity	An individual’s belief that he or she is better than others and deserves special rights and privileges.
	Insufficient self-control/self-discipline	Difficulty in developing sufficient self-control and frustration tolerance.
Other directedness	Subjugation	Feeling the need to submit to the control of others, in terms of needs and emotions, to avoid feelings of anger, revenge, or abandonment.
	Self-sacrifice	A focus on voluntarily satisfying others’ needs at the expense of one’s own.
	Approval seeking/recognition seeking	An exaggerated desire to gain recognition, respect or attention from or fit in with others, at the expense of developing a strong, authentic self.
Hyper vigilance/inhibition	Negativity/pessimism	Constantly focusing on the negative aspects of life (e.g., pain, death, loss, conflict, possible mistakes, etc.) while overlooking or downplaying the positive aspects.
	Emotional inhibition	The exaggerated inhibition of spontaneous actions, feelings and communication with others, in order to avoid disapproval, feelings of shame, or the loss of control.
	Unrelenting standards/hypercriticalness	Feeling the need to meet very high internalized standards of behavior and achievement, in order to avoid criticism.
	Punitiveness	A belief that people should be severely punished for their mistakes.

**Table 2 brainsci-13-01216-t002:** Description of the 14 schema modes included in four categories.

Domain	Schema Modes	Description
Maladaptive Child Modes	Vulnerable Child	The individual experiences a sense of unhappiness, anxiety, sadness, and helplessness.
	Angry Child	The individual experiences intense anger and even rage, and feels frustrated and impatient when his needs remain unmet.
	Enraged Child	The individual experiences extreme rage, leading to uncontrollable outbursts of aggression in which they may cause harm to others or destroy objects.
	Impulsive Child	The individual behaves selfishly under the influence of impulses or desires. It does not consider the consequences of its behavior and has difficulty deferring gratification.
	Undisciplined Child	The individual cannot force himself to complete routine, repetitive tasks because he quickly becomes frustrated and gives up.
Dysfunctional Coping Modes	Compliant Surrender	The individual is passive, subordinate, demands assurances and guarantees, and downplays his value for fear of conflict or rejection.
	Detached Protector	The individual avoids the mental pain associated with unmet needs by turning off all emotions, severing ties with others, and rejecting anyone’s help. He functions like a robot.
	Detached Self-Soother	The individual avoids experiencing emotions by engaging in activities that soothe, stimulate, or distract him (e.g., workaholism, gambling, extreme sports, casual sex, or drug use).
	Self-Aggrandizer	The individual strives for competition and power, behaves pretentiously, belittles and uses others to get what he wants. He shows superiority and expects special treatment.
	Bully and Attack	The individual uses threats, intimidation, and aggression to get what he wants or to protect himself from alleged harm.
Dysfunctional Parent Modes	Punitive Parent	That is the internalized voice of significant others criticizing or punishing the individual. It results in self-hatred, renunciation, self-harm, suicidal fantasies, and self-destructive behavior.
	Demanding Parent	It pressures and pushes the individual to meet exorbitant standards. It expects perfectionism, maintaining order and tidiness, striving for high status, high productivity, not wasting time.
Healthy Modes	Healthy Adult	Performs functions appropriate to adults, such as working, raising children, and taking responsibility. He also undertakes activities that are a source of pleasure, such as sex, and pursues intellectual, aesthetic, cultural interests, takes care of his health, and practices sports.
	Happy Child	The individual feels inner peace because his basic emotional needs are satisfied. He feels loved, fulfilled, competent, secure, praised, valuable, understood, resilient, optimistic, and spontaneous. He feels connected and cared for by others. At the same time, he has a sense of autonomy and control.

**Table 3 brainsci-13-01216-t003:** Characteristics of the studies.

Author/ Year/ Location	Objective	Sample Size	Characteristic of Participants	Assessment Measures	Results and Conclusions
Dutra et al., (2008), USA [27]	To examine the presence of EMS and investigate whether specific EMS might be related to other measures of trauma-related sequelae (in particular: to high suicide risk).	Total: 137	Patients seeking therapy for trauma survivors: 84% women, age: M = 38.3, SD = 11.2	– A modified version of the Young Schema Questionnaire-Short Form (YSQ-S) – Posttraumatic stress Diagnostic Scale (PDS) – The Dissociative Experiences Scale (DES) – The Self-harm and Risk Behaviors Questionnaire-Revised (SRBQ-R)	– The strongest correlation between dissociation and schemas of Social isolation/Alienation, Defectiveness/shame. – The strongest correlations between suicidal risk (suicidal ideation and the presence of a suicidal plan) and the same schemas plus schema of Failure. – No correlation between suicidality and self-harming behaviors with the EMS.
Dale et al., (2010), UK [28]	To explore the role of perceived parental bonding and EMSs in suicidal behavior.	Total: 156	Suicidal behavior group (n = 60): 73% women, age: M = 35.1, SD = 12.9. Comparison clinical group (n = 46): 60.4% women, age: M = 40.5. Non-clinical group (n = 48): 62.5% women, age: M = 39.9.	– The Parental Bonding Instrument (PBI) Shortened Version – Young Schema Questionnaire-Short Form (YSQ-S) – Beck Anxiety Inventory (BAI) – Beck Depression Inventory-Revised (BDI-II) – The Risk of Repetition Scale	– The suicidal behavior group scored significantly higher than the clinical and non-clinical groups on EMS, BAI, and BDI. – Significant positive correlations between the risk of repetition and the following EMS: Social alienation (a large effect), Defectiveness/Shame, Vulnerability to harm, Subjugation, Emotional inhibition, Entitlement, and Insufficient self-control. – Social alienation schema mediates between perceived parental care/control and risk of repetition. – Defectiveness/Shame schema mediates between perceived parental control and risk of repetition.
Leppänen et al., (2016), Finland [29]	To examine which EMS and schema modes emerged in parasuicidal and non-parasuicidal patients with borderline personality disorders (BPD).	Total: 60	All the patients with the diagnosis of BPD, 85% women, age: >20 (M = 32.4, SD = 8.6) Patients who fulfill criteria for parasuicidality (n = 46)	– The parasuicidal behavior subscale of the Borderline Personality Disorder Severity Index fourth version (BPDSI-IV) – The Young Schema Questionnaire (YSQ-L3a) – The Young Atkinson Mode Inventory (YAMI)	– Patients with parasuicidal behavior compared with patients without parasuicidal behavior: showed significantly higher scores in four EMSs: Emotional deprivation, Abandonment/Instability, Mistrust/Abuse, and Social isolation/Alienation. had significantly higher scores in four schema modes: the Vulnerable Child, the Angry Child, the Detached Protector, and the Compliant Surrender. had significantly lower scores in two schema modes: the Healthy Adult and the Happy Child. – Among the patients with parasuicidal behavior, EMSs correlated significantly with most of the schema modes; only a few significant correlations were identified among the patients without suicidal behavior.
Nilsson (2016), Denmark [30]	To compare patients with bipolar disorders (BD) with and without a history of suicide attempts in terms of EMSs.	Total: 49	Patients diagnosed with BD in current remission: With suicide attempts (n = 10): 80% women; age: M = 29.5, SD = 3.75 Without suicide attempts (n = 39): 71% women; age: M = 36.95, SD = 9.67	– Young Schema Questionnaire-Short Version (YSQ-S3)	– The BD patients with suicide attempts scored significantly higher than the BD patients without suicide attempts on three EMSs: Social isolation, Practical incompetence, and Entitlement.
Ahmadpanah et al., (2017), Iran [31]	To explore whether patients with major depressive disorders (MDD) and a history of suicide attempts differed in their EMSs from patients with MDD without such a history or from a healthy control.	Total: 90	Patients with MDD with a recent suicide attempt (n = 30): 70% women; age: M = 45.34, SD = 8.94. Patients with MDD without suicide attempts (n = 30): 63.3% women; age: M = 44.73, SD = 7.34. Gender- and age-matched healthy controls (n = 30): 60% females; age: M = 45.4, SD = 4.87	– The Young Schema Questionnaire-Long Form (YSQ-RE2R) – The Beck Depression Inventory (BDI) – Montgomery-Asberg Depression Rating Scale	– Patients with MDD, compared to healthy control, recorded higher scores for all the EMSs. – Compared to patients without suicide attempts and healthy controls, patients with suicidal attempts had higher scores in the following EMSs: Abandonment/Instability, Mistrust/Abuse, Emotional deprivation, Defectiveness/Shame, Social isolation/Alienation, Independence/Incompetence, Vulnerability to harm and illness, Failure, Emotional inhibition, Unrelenting standards/Hyper criticalness, and Punitiveness.
Khosravani et al., (2017), Iran [17]	To predict suicidal ideation and suicidal attempts in terms of EMSs and obsessive–compulsive (OC) symptom dimensions in patients diagnosed with obsessive–compulsive disorder (OCD).	Total: 60	Treatment-seeking outpatients with principal diagnoses of OCD: 51.7% of participants had lifetime suicide attempts, 75% had suicidal ideation; 51.7% women; age: M = 33.87, SD = 12.7	– The Scale for Suicide Ideation (SSI) – The Young Schema Questionnaire-Short Form (YSQ-SF) – The Yale-Brown Obsessive Compulsive Scale (Y-BOCS) – The Dimensional Obsessive–Compulsive Scale (DOCS) – The Depression Anxiety Stress Scales (DASS-21)	– Almost all EMSs (except for Abandonment, Unrelenting standards, and Entitlement) had significantly positive relations to suicidal ideation. – All OC symptom dimensions were positively correlated with suicidal ideation. – The schemas of Emotional deprivation and Mistrust/Abuse were associated with suicidal ideation (by controlling for depression, anxiety, and OCD severity). – The Mistrust/Abuse schema and the OC symptom dimension of unacceptable thoughts explained lifetime suicide attempts. – The Mistrust/Abuse schema, unacceptable thoughts, and depression significantly predicted suicidal ideation.
Langhinrichsen-Rohling et al., (2017), USA [32]	To determine whether specific maladaptive schemas mediate the relation between poor attachment and college students’ suicide proneness and ideation.	Total: 766	Students: 70% women; age: M = 19.9, SD = 3.7	– Life Attitudes Schedule-Short Form (LAS-SF) – Suicide Ideation Questionnaire (SIQ) – Inventory of Parent and Peer Attachment (IPPA) – Young Schema Questionnaire-Short Form (YSQ-3SF)	– Parental attachment was negatively related to suicide proneness, suicide ideation, and maladaptive self-schemas. – Parental attachment was negatively related to maladaptive schemas of abandonment, defectiveness, and emotional deprivation. – Schemas of abandonment and defectiveness were related to higher suicide proneness and suicide ideation. – Schemas of self-sacrifice and unrelenting standards were negatively related to suicide proneness and unrelated to suicide ideation. – Maladaptive schemas mediated the relation between parental attachment and suicide proneness and ideation.
Flink et al., (2017), Finland [19]	To explore underlying cognitive patterns associated with suicidal ideation by comparing EMSs among psychiatric outpatients with and without current suicidal ideation.	Total: 79	Psychiatric outpatients with major depressive disorder: 58.2% women; age: M = 40.43, SD = 11.73. 60.8% of them suffered from current suicidal ideation.	– The Young Schema Questionnaire Short Form-extended (YSQ-S2-extended) – The Beck Depression Inventory (BDI-21) – The Beck Hopelessness Scale (HS)	– Patients with suicidal ideation scored higher in the majority of EMSs than patients without suicidal ideation. – Vulnerability to harm or illness remained a predictor for suicidal ideation after controlling for the concurrent depressive symptom severity and hopelessness.
Frias et al., (2018), Spain [33]	To determine if any schema domains were directly associated with dimensional psychopathology (e.g., with suicidal ideation) in patients with borderline personality disorders (BPD). To clarify if there was a direct association of any schema domains with categorical psychopathology in patients with BPD.	Total: 102	Patients with BPD: 91.2% women; age: M = 35.99, SD = 11.9	– The Scale for Suicidal Ideation (SSI) – The Buss and Perry Aggression Questionnaire (AQ) – The Symptom CheckList-Revised (SCL-90-R) – The Young Schema Questionnaire-Short Form (YSQ-SF) – The Clinical Global Impression scale	– BPD patients had middle-high or high scores in all EMSs and schema domains. – Disconnection/Rejection significantly and positively predicted: the severity of suicidal ideation, physical aggression and hostility, dimensions of general psychopathology like depression, psychoticism, hostility, and paranoid ideation, the presence of lifetime eating disorders and posttraumatic stress disorder comorbidity. – Other-directedness: significantly and positively predicted verbal aggression and anger, somatization, obsessive-compulsive symptoms, anxiety, phobic anxiety, hostility, and interpersonal sensitivity significantly predicted the absence of lifetime substance use disorders.
Khosravani et al., (2019), Iran [18]	To assess the associations of EMSs and clinical factors (hypomanic/manic and depressive symptoms) with suicidal risk.	Total: 100	Patients with bipolar disorders (BP) in remission: 59% of patients had lifetime suicide attempts; 59% showed high suicidal risk; 43% women; age: M = 36.7, SD = 8.5	– The Young Schema Questionnaire-Short Form (YSQ-SF) – The Bipolar Depression Rating Scale (BDRS) – The Young Mania Rating Scale (YMRS) – The Beck Scale for Suicidal Ideation (BSSI)	– BP patients with lifetime suicidal attempts had higher scores on the Entitlement and Social isolation schemas, depression, and hypomanic/manic symptoms than those without such attempts. – BP patients with high suicidal risk had higher levels of depressive and hypomanic/manic symptoms and some EMSs than those without high suicidal risk. – Hypomanic/manic symptoms and Entitlement and Defectiveness schemas were associated with lifetime suicide attempts.
Azadi et al., (2019), Iran [16]	To investigate the associations of EMSs and clinical factors with suicidal risk among patients with schizophrenia.	Total: 82	Patients with a principal diagnosis of schizophrenia: 41.5% had lifetime suicide attempts; 42.7% were at high current suicidal risk; 58.5% women; age: M = 34.78, SD = 9.1	– The Young Schema Questionnaire-Short Form (YSQ-SF) – The Beck Depression Inventory-II (BDI-II) – The Beck Scale for Suicide Ideation (BSSI) – The Positive and Negative Syndrome Scale (PANSS)	– Individuals with schizophrenia who had attempted suicide (compared to those without) had significantly higher EMSs, current suicidal ideation, and a family history of suicide attempts. – The schema of Emotional deprivation, positive symptoms, and depression were significantly associated with current suicidal ideation. – The schema of Emotional deprivation was significantly associated with lifetime suicide attempts.
Ha et al., (2022), Republic of Korea [34]	To examine the application of interpersonal-psychological theory and EMS to suicidal ideation and suicide attempts in South Korean university students.	Total: 367	University students: 79% women, age: M = 23.38, SD = 3.56	– The Interpersonal Needs Questionnaire’s Korean version (K-INQ) – The Young Schema Questionnaire-Short Form (YSQ-SF) – The Suicide Ideation Scale (SIS) – The Acquired Capability for Suicide Scale	– The interpersonal needs and EMS were significantly associated with suicidal ideation and influencing suicide attempts. – The acquired capability for suicide moderated the relationship between suicidal ideation and attempts.

**Table 4 brainsci-13-01216-t004:** The main results of the studies: associations between EMSs and suicidality taking into account the measure of suicidal risk and controlled variables.

Early Maladaptive Schema	Dutra et al. (2008) [27]	Dale et al. (2010) [28]	Leppänen et al. (2016) ^1^ [29]	Nilsson (2016) ^2^ [30]	Ahmadpanah et al. (2017) ^3^ [31]	Khosravani et al. (2017) ^4^ [17]	Langhinrichsen-Rohling et al. (2017) [32]	Flink et al. (2017) ^3^ [19]	Frias et al. (2018) ^1^ [33]	Khosravani et al. (2019) ^2^ [18]	Azadi et al. (2019) ^5^ [16]	Ha et al. (2022) ^6^ [34]
1. Disconnection/Rejection												
Abandonment /Instability			p/a *		A ***		pr/i		i *	a **/r ** a **	i(d) *	i/a ***
Mistrust /Abuse			p/a *		a***	i(d,a,o) ** a ***		i **		a ***/r ***	i(d) *	
Emotional Deprivation			p/a *		a***	i(d,a,o) **		i **		a **/r(d,m) **	i(d)/a **	
Defectiveness /Shame	i **/p **/a *	rr **			a ***	i **	pr/i	i ***		a***/r(d,m) *** i **	i(d) *	
Social isolation /Alienation	**/p **	rr ***	p/a *	a *	a ***	i **		i ***		a(d,m) ** /r(d,m) ** a **/i **	i(d) *	
2. Impaired autonomy/performance												
Dependence /Incompetence	i */p **			a *	a ***	i **		i ***		a **/r(d,m) * a **	i ***	i/a ***
Vulnerability to harm and illness	n/a	rr **			a **	i **		i(d,h) ***		a ***/r(d,m) **	i ***	
Enmeshment /Undeveloped self						i **				a ***/r ***	i ***	
Failure	i **/p **/a *				a ***	i **		i *		a ***/r(d,m) ** i **	i ***	
3. Impaired limits												
Entitlement /Grandiosity	n/a	rr *		a *						a(d,m) *** /r(d,m) *** a **/i **	i ***	i/a ***
Insufficient self-control /self-discipline	n/a	rr **				i **				a ***/r ***	i ***	
4. Other directedness												
Subjugation		rr **				i *		i *		a ***/r(d,m) **	i ***	i/a ***
Self-sacrifice						i **	pr-	i *		a ***/r *		
Approval seeking /Recognition seeking	n/a	n/a				n/a	n/a		n/a	n/a	n/a	n/a
5. Hyper vigilance/Inhibition												
Negativity/ Pessimism	n/a	n/a				n/a	n/a	i **	n/a	n/a	n/a	n/a
Emotional inhibition		rr *			a ***	i **				a **/r(d,m) ** i **	i ***	i/a ***
Unrelenting standards /hypercriticalness					a ***		pr-			a **/r * a **		
Punitiveness	n/a	n/a			a ***	n/a	n/a	i **	n/a	n/a	n/a	n/a

Note. gray background = significant results; empty cells = no significant relationship; “n/a” = not applicable (EMS was not measured); “i” = positive correlation with current suicidal ideation; “p” = current suicidal plan; “rr” = risk of repetition; “pr” = suicide proneness; “pr-” = negative correlation with suicide proneness; “a” = suicidal attempt in history; “i(d,a,o,h,m)” = correlation with suicidal ideation after controlling for depression, anxiety, OCD severity, hopelessness, and manic symptoms; * *p* < 0.05; ** *p* < 0.01; *** *p* < 0.001. ^1^—patients diagnosed with borderline personality disorders; ^2^—patients with bipolar disorder; ^3^—patients with major depressive disorders; ^4^—patients with OCD; ^5^—patients with schizophrenia; ^6^— students.

## Data Availability

Data sharing is not applicable for this article as no new data were created in this study.

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
