# Peer review of "Early Maladaptive Schemas and Schema Modes among People with Histories of Suicidality and the Possibility of a Universal Pattern: A Systematic Review"

_brainsci, 2023, doi:10.3390/brainsci13081216_

Round 1

Reviewer 1 Report

This paper is a systematic review on the relation between EMS/Schema Modes and different measures of suicidality. It is a relevant topic and a carefully executed systematic review, using the PRISMA guidelines. 

I have one major concern on the abstract and discussion: 

It was mentioned very correctly that correlations do not mean causality, but in the abstract a suggestion was made that identification of the schemas and modes provides an opportunity for practitioners to apply appropriate treatment techniques (line 21 and 22). And again, in the discussion, in lines 621 to 624 a statement is made about intervention that is not based on the results of the present study and should be phrased with more caution. Please rephrase more carefully. 

I have a couple of minor concerns: 

  1. 1. In the introduction I do not understand the metaphor used in line 51 to 54 of baggage. Please clarify or remove the metaphor. 

  1. 2. In Figure 1 the studies excluded based on the titles add up to 28 not 27. But the difference between 42 studies and 15 is 27, not 28. Please clarify. 

  1. 3. In Figure 1 the studies excluded based on abstracts add up to 3, not 2. Please clarify 

  1. 4. In Table 3 please also mention the numbers of the article references 

  1. 5. Table 4 is not clear with all the abbreviations in the cells of the table. Is there a way to make this table easier to read? And what is the difference between n/a and an empty cell? 

  1. 6. In the headings 3.4.1 the word patients is used, whilst there are two studies that use students as participants, not patients. Please adjust the heading. 

  1. 7. In line 538 the words more than a dozen is used, whilst it is exactly a dozen, please replace by ‘… based on 12 studies’. 

  1. 8. Are all the supplementary files necessary? Can they be combined?

The quality of the English languages is high throughout the manuscript. 

Only one remark: In the abstract, line 15 and 16: ‘the relationship of EMSs and Schema Modes to measures ....’ Please use ‘the relationship between A and B’, not ‘of A to B’.

Reviewer 2 Report

-Was the status of publication (i.e. grey literature) used as an inclusion criterion? The authors should state that they searched for reports regardless of their publication type.

- Authors need to clarify why they decided to include participants from 16 years of age and exclude those over 70 years of age
